# Long-Term Symptoms among Hospitalized COVID-19 Patients 48 Weeks after Discharge—A Prospective Cohort Study

**DOI:** 10.3390/jcm10225298

**Published:** 2021-11-15

**Authors:** Martin Mølhave, Steffen Leth, Jesper Damsgaard Gunst, Søren Jensen-Fangel, Lars Østergaard, Christian Wejse, Jane Agergaard

**Affiliations:** 1Department of Infectious Diseases, Aarhus University Hospital, 8200 Aarhus, Denmark; stefleth@rm.dk (S.L.); jesdam@rm.dk (J.D.G.); soejense@rm.dk (S.J.-F.); larsoest@rm.dk (L.Ø.); wejse@ph.au.dk (C.W.); janeager@rm.dk (J.A.); 2Department of Medicine, Regional Hospital Unit West Jutland, 7400 Herning, Denmark

**Keywords:** COVID-19, long-term symptoms, long-term complications, prospective studies, hospitalized patients

## Abstract

Follow-up studies of COVID-19 survivors have been performed to characterize persistence of long-term symptoms, but data are scarce on one year of follow-up. This study provides data from 48 weeks of follow-up after discharge. All patients discharged from the Department of Infectious Diseases at Aarhus University Hospital, Denmark between 1 March and 1 July 2020 were followed for 48 weeks. In total, 45 of 66 eligible patients were interviewed after 48 weeks. The median age was 57 (IQR 51–70) years, the majority were female (53%) and Caucasian (87%). Median BMI was 28.1 (IQR 24.8–32.6) kg/m^2^. One or more comorbidities were registered among 62% of the patients. In total, 39 out of 45 (87%) interviewed patients reported persistence of at least one symptom 48 weeks after hospitalization with COVID-19. Most frequently reported symptoms were fatigue, dyspnea, and concentration difficulties. This study provides new long-term data following COVID-19, contributing to the accumulating data of COVID-19 sequelae. Many patients suffer long-term sequelae and further research is urgently needed to gain further knowledge of the duration and therapeutic options.

## 1. Introduction

Coronavirus disease 19 (COVID-19) has led to staggering conditions including lockdowns and hospitals filled with patients. Furthermore, a substantial number of patients present with long-term symptoms after COVID-19 [1]. Several studies have addressed and characterized persistent symptoms following the acute phase of COVID-19. A new study by Huang et al. [2] characterizing persistent symptoms among 1276 patients six months and one year after hospitalization with acute COVID-19 found that 68% and 49% reported at least one symptom after six months and one year, respectively. The most frequently reported symptoms were fatigue or muscle weakness, pain or discomfort, sleep difficulties, and dyspnea. Another study by Augustin et al. [3] found that 35% of hospitalized patients experienced fatigue, shortness of breath, anosmia, or ageusia seven months after the acute phase of COVID-19. Xiong et al. [4] showed that half of patients had at least one symptom at three months follow-up. Other studies have shown similar outcomes among patients reporting symptoms following COVID-19; fatigue, in particular, was frequently reported [5,6,7]. Taking into consideration the severity of long-term symptoms [8], it is essential to characterize and determine the duration of long-term symptoms following COVID-19. In this report, we describe the prevalence of persistent symptoms 48 weeks after hospitalization with COVID-19 in a cohort of adult patients.

## 2. Materials and Methods

Participants were admitted due to COVID-19 at Department of Infectious Diseases at Aarhus University Hospital, Denmark between 1 March and 1 July 2020. From part of this cohort, clinical data regarding the patients’ symptoms during admission as well as a 6- and 12-week follow-up have previously been described by Leth et al. [9]. Patients had tested positive for SARS-CoV-2 with a polymerase chain reaction (PCR) test, were more than 18 years of age, and hospitalized for at least 12 h. Patient data regarding age, sex, ethnicity, BMI, smoking status, and comorbidities were registered at admission.

In this study, patients were contacted by telephone and interviewed 48 weeks after discharge. Patients were systematically asked to identify the presence and duration of symptoms using a clinical interview guide (Figure A1). The severity of dyspnea was evaluated using the Medical Research Council (MRC) score (3–5 being severe dyspnea), and cognitive function was assessed by the validated Orientation-Memory-Concentration (OMC) test (24 or lower being impaired) (Figure A1). The same questionnaire was used for the 6- and 12-week follow-up in the study by Leth et al. [9].

According to Danish law, register- and questionnaire-based studies do not require ethics approval (Danish Committee Act, Section 14, Subsection 2). The same applied for the Central Denmark Region Committee on Health Research Ethics (reference 1-10-72-181-20). The data collection and interview study were approved by the Central Denmark Region (references 1-45-70-5-20 and 1-16-02-4-21). Oral and written consent were obtained from all participants.

Data were analyzed using Stata Intercooled version 11. Multivariate analysis by logistical regression was used to calculate the odds ratio (OR) for risk factors of the most frequent symptoms and ICU admission. *p*-values below 0.05 were considered statistically significant. Risk factors tested for association with the most common persistent symptoms were age, sex, BMI, smoking, and comorbidities.

## 3. Results

Between 1 March and 1 July 2020, 78 patients were admitted to the Department of Infectious Diseases. Eleven patients died during or after admission and one patient was unable to give consent due to dementia. A total of 66 patients were eligible, of whom two patients were lost to follow-up due to travel, 16 patients had not provided consent to participate, and three patients did not answer phone calls for interviews. Hence, a total of 45 patients were interviewed 48 weeks after discharge. A flowchart of exclusion and inclusion of patients can be seen in Figure A2.

Patients were interviewed between 19 February and 1 July 2021, at a median of 350 (IQR 341–380) days after discharge. Median duration from symptom onset until hospital admission was eight (IQR 5–10) days, median duration of hospitalization was seven (IQR 3–12) days; and 18% of patients received ICU care during admission.

The median age of the 45 participants at the time of the interview was 57 (IQR 51–70) years; the majority were female (53%) and Caucasian (87%). Median BMI was 28.1 (IQR 24.8–32.6) kg/m^2^, 40% were active or previous smokers, 73% received oxygen therapy during admission, and 62% had one or more known comorbidities with hypertension (29%) being the most prevalent, followed by asthma (9%), coronary heart disease (9%), and any type of malignancy (9%). No patients received treatment with Remdesivir or steroids due to COVID-19. Eight patients received steroids on indications other than COVID-19. Table A1 summarizes baseline characteristics of participants.

A fully comprehensive table of all reported symptoms can be seen in Table A2. New symptoms or a worsening of a symptom after acute COVID-19 were registered. The most frequent symptoms were fatigue, dyspnea, and concentration difficulties, reported by 60% (27/45), 40% (18/45), and 40% (18/45), respectively. Symptoms that could refer to multiple organs were reported: headache, altered bowel habits, impaired sense of smell, cough, myalgia (Table A2). In total, 24% scored ≤24 on the OMC test. The proportion of patients reporting at least one symptom 48 weeks after discharge was 87%.

ORs for fatigue, dyspnea, and concentration difficulties were calculated based on risk factors using multivariate logistical regression based on patient characteristics: BMI ≥25 vs. <25 kg/m^2^, age ≥60 vs. <60 years, smoker or previous smoker vs. non-smoker, male vs. female, comorbidities vs. no comorbidities, and receiving oxygen therapy vs. not receiving oxygen therapy. These factors have been associated with a more severe clinical outcome during the acute and post-acute phase of COVID-19 [1,2,6,8,10]. People of an ethnicity other than Caucasian had a reduced risk of experiencing fatigue (OR 0.061, CI (0.0051–0.72), *p* = 0.027) at 48 weeks after discharge. Female sex showed a trend, though not statistically significant, to increased odds for fatigue and concentrating difficulties. All other associations showed non-significant results. Table A3 shows the results of these analyses.

## 4. Discussion

Data from this study showed the presence of long-term persistent symptoms after COVID-19 almost one year after hospitalization with COVID-19. One or more symptoms were reported by 87% of patients, and fatigue, dyspnea, and difficulties concentrating were still present in a substantial number of patients 48 weeks after discharge.

Our data indicate a significant presence of long-lasting symptoms. Previously published studies have found between 28% and 68% of patients reporting at least one symptom at 10 weeks to 12 months after hospitalization in the acute phase of COVID-19 [2,3,4,7,11].

Huang et al. [2] reported one or more sequelae symptoms in 68% at six months following discharge, which later dropped to 49% of patients one year after discharge. Half of the patients (1276 of 2469) discharged alive took part in the study. In our cohort, Leth et al. [9] found that 86% and 96% reported persistence of at least one symptom after six and 12 weeks, respectively. In the present study, we evaluated 45 of 66 patients discharged alive (Figure A2) and found 87% with one or more symptoms.

In line with several other studies [2,3,4,5,7,9,11], fatigue was reported as the most frequent symptom (60%) in our study. Interestingly, Townsend et al. [6] found no association between the severity of COVID-19 and the presence of fatigue following COVID-19. Mechanisms behind fatigue experienced after COVID-19 are yet unknown. However, a study by Agergaard et al. [12] found in a cohort of 20 patients that 11 patients with myopathy all experienced fatigue. The patients’ myopathy diagnosis was verified by a myopathic quantitative electromyography (qEMG). Based on these findings, the authors suggest that myopathy may be an important cause of fatigue in post-COVID-19 patients. If myalgia has an effect on fatigue experienced by COVID-19 patients, it could explain why myalgia or muscle weakness was among their most frequently reported symptoms in our and other studies [2,5].

As described in our study, dyspnea [2,3,5,7] and concentration difficulties [8] are also among the most frequently reported persistent symptoms in other studies. Among the self-reporting non-hospitalized patients, Ayoubkhani and Pawelek [13] estimated that one million people living in the UK experienced symptoms at least four weeks after suspected coronavirus infection; the number was 376,000 after one year. Fatigue, shortness of breath, and concentration difficulties were also the most common symptoms, indicating similarities between hospitalized and non-hospitalized individuals.

A pathophysiological explanation for the development of persistent symptoms has not been established. Possible mechanisms of post-acute COVID-19 may include virus-specific pathophysiological traits, immunological disruptions, and damage following the acute phase of COVID-19 as well as sequelae following critical illness [8].

In the present study, multivariate analyses between the three most commonly reported symptoms and various risk factors proved to be insignificant, apart from people of an ethnicity other than Caucasian having a reduced risk of experiencing fatigue (OR 0.061, CI (0.0051–0.72), *p* = 0.027) 48 weeks after discharge. To our knowledge, it has not been reported in other studies that people of an ethnicity other than Caucasian have a lower risk of experiencing long-term symptoms after COVID-19. The lower odds for fatigue may merely reflect that patients with another ethnic origin experiencing fatigue were more likely than others to decline the invitation to take part in the study due to language barriers. In a study by Yomogida et al., Black/African American participants had higher odds of reporting dyspnea and myalgia/arthralgia compared with other racial/ethnic groups at two months follow-up after a positive COVID-19 test result [14]. When considering this contradictory result in light of our results regarding different long-term symptoms as well as the small number of patients in our cohort, this result may not be of any importance to COVID-19 patients in general.

We also found a non-significant tendency for female sex to be at a greater risk of experiencing concentration difficulties (OR 4.06, CI (0.89–18.48), *p* = 0.07) and fatigue (OR 3.65, CI (0.84–15.92), *p* = 0.08) 48 weeks after discharge compared with males. Townsend et al. and Huang et al. also found an association between female sex and fatigue [2,6]. A possible cause of this observation could be that women are more likely to seek a doctor when experiencing a symptom [15] or due to sex differences in immune responses against COVID-19, as suggested by Takahashi et al. [16]. The fact that we did not find other significant outcomes in our analysis is most likely due to the small cohort size and the heterogenic nature of long-term symptoms after COVID-19.

The small size of the cohort is the main weakness of this study, which makes it difficult to draw any conclusions based on these data alone. It is also a limitation that most symptoms reported were not graded for severity and based on the patients’ personal experience, which could cause bias to either over- or under-report symptoms. Another limitation is that none of these symptoms are specific for acute or long-term COVID-19, which makes it difficult to conclude that every symptom reported in this study is directly correlated with COVID-19 as they could be caused by other illnesses. Furthermore, it is likely that patients who have had the most successful recovery would be the least motivated to participate in a follow-up study.

Assuming the 21 non-participants were all asymptomatic, the proportion of patients reporting at least one symptom would be 59% (39/66) instead of 87%. This would change our results to be more similar to the one-year follow-up by Huang et al., where 68% and 49% of discharged patients reported at least one symptom after six months and one year, respectively.

## 5. Conclusions

At 48 weeks after discharge of patients with COVID-19, 87% patients reported persistence of at least one symptom. In accordance with other studies investigating COVID-19 sequelae, the most frequent symptoms were fatigue, dyspnea, and concentration difficulties. These results indicate long-lasting symptoms after COVID-19 and that long follow-up studies in large populations and multidisciplinary investigations are needed to fully elucidate the consequences.

## Data Availability

Data used for this paper are stored in a REDCap database requiring a two-factor validation for log-in.

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
