# Peer review of "Long-Term Symptoms among Hospitalized COVID-19 Patients 48 Weeks after Discharge—A Prospective Cohort Study"

_jcm, 2021, doi:10.3390/jcm10225298_

Round 1

Reviewer 1 Report

The present manuscript reported long-term symptoms among COVID-19 patients. The authors conducted a prospective cohort study to elucidate the long-term symptoms among COVID-19 survivors. I read the manuscript with great interest. The manuscript is well written, and the data are valuable because the sequelae of COVID-19 have become a worldwide health problem.

  1. As the authors mentioned, females tend to have more symptoms. It would be helpful if the authors suggest the mechanisms why did female COVID-19 survivors tend to experience long-term symptoms.
  2. Was there any association between COVID-19 treatment and long-term symptoms?

Reviewer 2 Report

The authors of “Long-term symptoms among hospitalized COVID-19 patients 48 weeks after discharge – a prospective cohort study” have presented a study where 45 patients admitted to an infectious diseases unit for COVID-19 are followed up almost a year after discharge and assessed using a telephone questionnaire for a range of symptoms. The study adds to the literature around long COVID-19 and lingering symptoms found in patients following COVID-19 related hospitalization. However, there are some areas that should be addressed before publication; these are identified and discussed below.

Main issues

1) There are a number of grammatical errors in the manuscript. Some of these are described in Minor issues below, but they need to be fixed prior to publication.

2) This was a very small study of only 45 participants. This is somewhat acknowledged in the Discussion, but it definitely needs to be reinforced as the primary weakness and makes it challenging to draw concrete conclusions

3) The questionnaire was not done at baseline (which could be defined as at discharge, potentially) or at any additional time point besides the 48 weeks post-discharge time point. This makes it challenging to consider the reported symptoms as objectively new symptoms.

4) For the risk factors tested for association with the most common persisting symptoms, were these determined at the 48 week time point or at the time of acute COVID-19 infection? Comorbidities, BMI, and smoking status might change during the 48 weeks and it would be helpful to just have a statement that explains when these risk factors were identified in the study’s timeline (eg, at admission, discharge, or the 48 week time point).

5) It is also important to highlight that it cannot be concluded that the new symptoms are at all related to having been diagnosed with COVID-19 previously. 48 weeks is a long time and a number of new medical issues may have cropped up in that time period unrelated to prior COVID-19. For example, it is unclear whether the presence of fatigue at 48 weeks is related to COVID-19 or a diagnosis of some other medical condition that has presented in the interim. This is further exacerbated by the cohort being so small.

6) The association of certain symptoms with female sex was not statistically significant – it is important to be more explicit in the text that it did not reach statistical significance (particularly in the text at lines 106-107 and 152-154).

7) Please provide some explanation for the biological plausibility for some of the findings of the study (ie, that non-Caucasians seemed to have less fatigue and that females had a tendency toward fatigue and difficulty concentrating). If there is none, then it would be helpful to note that.

Minor issues

Line 20: insert “the” between “between” and “1st of March”

Lines 36-37: “…since the first tendencies of persisting symptoms emerged” is very unclear. Please modify it so that it makes more sense.

Line 56: replace “were” with “had"

Line 57: add “of age” after “18 years”

Line 64: change “ethical” to “ethics”

Line 73: add “and” before “comorbidities”

Line 76: add “and” after “admission”

Line 96: it is not entirely clear what “fluctuations in stool” means. Does it refer to altered bowel habits?

Line 104: the first letter of “people” needs capitalization at the beginning of the sentence

Line 105: change “having” to “had”

Line 159: change “after” to “for”

Lines 163-164: the final sentence of the Discussion is just an observation and seems to need a bit more of a conclusion, such as a judgement as to the significance of the results if only 59% of the patients had ongoing symptoms.
